# AN IMAGE IS WORTH 16x16 WORDS:
# TRANSFORMERS FOR IMAGE RECOGNITION AT SCALE

**Alexey Dosovitskiy**[*,†]**, Lucas Beyer**[*]**, Alexander Kolesnikov**[*]**, Dirk Weissenborn**[*]**,**
**Xiaohua Zhai**[*]**, Thomas Unterthiner, Mostafa Dehghani, Matthias Minderer,**
**Georg Heigold, Sylvain Gelly, Jakob Uszkoreit, Neil Houlsby**[*,†]

[*]equal technical contribution, [†]equal advising
Google Research, Brain Team
{adosovitskiy, neilhoulsby}@google.com

## ABSTRACT

While the Transformer architecture has become the de-facto standard for natural language processing tasks, its applications to computer vision remain limited. In vision, attention is either applied in conjunction with convolutional networks, or used to replace certain components of convolutional networks while keeping their overall structure in place. We show that this reliance on CNNs is not necessary and a pure transformer applied directly to sequences of image patches can perform very well on image classification tasks. When pre-trained on large amounts of data and transferred to multiple mid-sized or small image recognition benchmarks (ImageNet, CIFAR-100, VTAB, etc.), Vision Transformer (ViT) attains excellent results compared to state-of-the-art convolutional networks while requiring substantially fewer computational resources to train.[1]

## 1 INTRODUCTION

Self-attention-based architectures, in particular Transformers (Vaswani et al., 2017), have become the model of choice in natural language processing (NLP). The dominant approach is to pre-train on a large text corpus and then fine-tune on a smaller task-specific dataset (Devlin et al., 2019). Thanks to Transformers' computational efficiency and scalability, it has become possible to train models of unprecedented size, with over 100B parameters (Brown et al., 2020; Lepikhin et al., 2020). With the models and datasets growing, there is still no sign of saturating performance.

In computer vision, however, convolutional architectures remain dominant (LeCun et al., 1989; Krizhevsky et al., 2012; He et al., 2016). Inspired by NLP successes, multiple works try combining CNN-like architectures with self-attention (Wang et al., 2018; Carion et al., 2020), some replacing the convolutions entirely (Ramachandran et al., 2019; Wang et al., 2020a). The latter models, while theoretically efficient, have not yet been scaled effectively on modern hardware accelerators due to the use of specialized attention patterns. Therefore, in large-scale image recognition, classic ResNet-like architectures are still state of the art (Mahajan et al., 2018; Xie et al., 2020; Kolesnikov et al., 2020).

Inspired by the Transformer scaling successes in NLP, we experiment with applying a standard Transformer directly to images, with the fewest possible modifications. To do so, we split an image into patches and provide the sequence of linear embeddings of these patches as an input to a Transformer. Image patches are treated the same way as tokens (words) in an NLP application. We train the model on image classification in supervised fashion.

When trained on mid-sized datasets such as ImageNet without strong regularization, these models yield modest accuracies of a few percentage points below ResNets of comparable size. This seemingly discouraging outcome may be expected: Transformers lack some of the inductive biases

---

[1]Fine-tuning code and pre-trained models are available at https://github.com/google-research/vision_transformer

inherent to CNNs, such as translation equivariance and locality, and therefore do not generalize well when trained on insufficient amounts of data.

However, the picture changes if the models are trained on larger datasets (14M-300M images). We find that large scale training trumps inductive bias. Our Vision Transformer (ViT) attains excellent results when pre-trained at sufficient scale and transferred to tasks with fewer datapoints. When pre-trained on the public ImageNet-21k dataset or the in-house JFT-300M dataset, ViT approaches or beats state of the art on multiple image recognition benchmarks. In particular, the best model reaches the accuracy of $88.55\%$ on ImageNet, $90.72\%$ on ImageNet-ReaL, $94.55\%$ on CIFAR-100, and $77.63\%$ on the VTAB suite of 19 tasks.

## 2 RELATED WORK

Transformers were proposed by Vaswani et al. (2017) for machine translation, and have since become the state of the art method in many NLP tasks. Large Transformer-based models are often pre-trained on large corpora and then fine-tuned for the task at hand: BERT (Devlin et al., 2019) uses a denoising self-supervised pre-training task, while the GPT line of work uses language modeling as its pre-training task (Radford et al., 2018; 2019; Brown et al., 2020).

Naive application of self-attention to images would require that each pixel attends to every other pixel. With quadratic cost in the number of pixels, this does not scale to realistic input sizes. Thus, to apply Transformers in the context of image processing, several approximations have been tried in the past. Parmar et al. (2018) applied the self-attention only in local neighborhoods for each query pixel instead of globally. Such local multi-head dot-product self attention blocks can completely replace convolutions (Hu et al., 2019; Ramachandran et al., 2019; Zhao et al., 2020). In a different line of work, Sparse Transformers (Child et al., 2019) employ scalable approximations to global self-attention in order to be applicable to images. An alternative way to scale attention is to apply it in blocks of varying sizes (Weissenborn et al., 2019), in the extreme case only along individual axes (Ho et al., 2019; Wang et al., 2020a). Many of these specialized attention architectures demonstrate promising results on computer vision tasks, but require complex engineering to be implemented efficiently on hardware accelerators.

Most related to ours is the model of Cordonnier et al. (2020), which extracts patches of size $2 \times 2$ from the input image and applies full self-attention on top. This model is very similar to ViT, but our work goes further to demonstrate that large scale pre-training makes vanilla transformers competitive with (or even better than) state-of-the-art CNNs. Moreover, Cordonnier et al. (2020) use a small patch size of $2 \times 2$ pixels, which makes the model applicable only to small-resolution images, while we handle medium-resolution images as well.

There has also been a lot of interest in combining convolutional neural networks (CNNs) with forms of self-attention, e.g. by augmenting feature maps for image classification (Bello et al., 2019) or by further processing the output of a CNN using self-attention, e.g. for object detection (Hu et al., 2018; Carion et al., 2020), video processing (Wang et al., 2018; Sun et al., 2019), image classification (Wu et al., 2020), unsupervised object discovery (Locatello et al., 2020), or unified text-vision tasks (Chen et al., 2020c; Lu et al., 2019; Li et al., 2019).

Another recent related model is image GPT (iGPT) (Chen et al., 2020a), which applies Transformers to image pixels after reducing image resolution and color space. The model is trained in an unsupervised fashion as a generative model, and the resulting representation can then be fine-tuned or probed linearly for classification performance, achieving a maximal accuracy of 72% on ImageNet.

Our work adds to the increasing collection of papers that explore image recognition at larger scales than the standard ImageNet dataset. The use of additional data sources allows to achieve state-of-the-art results on standard benchmarks (Mahajan et al., 2018; Touvron et al., 2019; Xie et al., 2020). Moreover, Sun et al. (2017) study how CNN performance scales with dataset size, and Kolesnikov et al. (2020); Djolonga et al. (2020) perform an empirical exploration of CNN transfer learning from large scale datasets such as ImageNet-21k and JFT-300M. We focus on these two latter datasets as well, but train Transformers instead of ResNet-based models used in prior works.

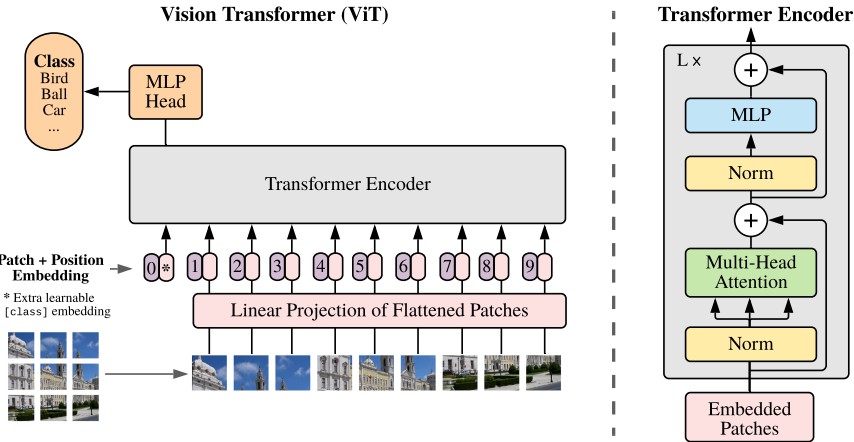

Figure 1: Model overview. We split an image into fixed-size patches, linearly embed each of them, add position embeddings, and feed the resulting sequence of vectors to a standard Transformer encoder. In order to perform classification, we use the standard approach of adding an extra learnable "classification token" to the sequence. The illustration of the Transformer encoder was inspired by Vaswani et al. (2017).

## 3 METHOD

In model design we follow the original Transformer (Vaswani et al., 2017) as closely as possible. An advantage of this intentionally simple setup is that scalable NLP Transformer architectures – and their efficient implementations – can be used almost out of the box.

### 3.1 VISION TRANSFORMER (VIT)

An overview of the model is depicted in Figure 1. The standard Transformer receives as input a 1D sequence of token embeddings. To handle 2D images, we reshape the image $\mathbf{x} \in \mathbb{R}^{H \times W \times C}$ into a sequence of flattened 2D patches $\mathbf{x}_p \in \mathbb{R}^{N \times (P^2 \cdot C)}$, where $(H, W)$ is the resolution of the original image, $C$ is the number of channels, $(P, P)$ is the resolution of each image patch, and $N = HW/P^2$ is the resulting number of patches, which also serves as the effective input sequence length for the Transformer. The Transformer uses constant latent vector size $D$ through all of its layers, so we flatten the patches and map to $D$ dimensions with a trainable linear projection (Eq. 1). We refer to the output of this projection as the patch embeddings.

Similar to BERT's [class] token, we prepend a learnable embedding to the sequence of embedded patches ($\mathbf{z}_0^0 = \mathbf{x}_{\text{class}}$), whose state at the output of the Transformer encoder ($\mathbf{z}_L^0$) serves as the image representation $\mathbf{y}$ (Eq. 4). Both during pre-training and fine-tuning, a classification head is attached to $\mathbf{z}_L^0$. The classification head is implemented by a MLP with one hidden layer at pre-training time and by a single linear layer at fine-tuning time.

Position embeddings are added to the patch embeddings to retain positional information. We use standard learnable 1D position embeddings, since we have not observed significant performance gains from using more advanced 2D-aware position embeddings (Appendix D.3). The resulting sequence of embedding vectors serves as input to the encoder.

The Transformer encoder (Vaswani et al., 2017) consists of alternating layers of multiheaded self-attention (MSA, see Appendix A) and MLP blocks (Eq. 2, 3). Layernorm (LN) is applied before every block, and residual connections after every block (Wang et al., 2019; Baevski & Auli, 2019).

The MLP contains two layers with a GELU non-linearity.

$$\mathbf{z}_0 = [\mathbf{x}_{\text{class}}; \mathbf{x}_p^1 \mathbf{E}; \mathbf{x}_p^2 \mathbf{E}; \cdots; \mathbf{x}_p^N \mathbf{E}] + \mathbf{E}_{pos}, \qquad \mathbf{E} \in \mathbb{R}^{(P^2 \cdot C) \times D}, \ \mathbf{E}_{pos} \in \mathbb{R}^{(N+1) \times D} \qquad (1)$$

$$\mathbf{z'}_\ell = \text{MSA}(\text{LN}(\mathbf{z}_{\ell-1})) + \mathbf{z}_{\ell-1}, \qquad\qquad \ell = 1 \ldots L \qquad\qquad\qquad (2)$$

$$\mathbf{z}_\ell = \text{MLP}(\text{LN}(\mathbf{z'}_\ell)) + \mathbf{z'}_\ell, \qquad\qquad \ell = 1 \ldots L \qquad\qquad\qquad (3)$$

$$\mathbf{y} = \text{LN}(\mathbf{z}_L^0) \qquad\qquad\qquad\qquad\qquad\qquad\qquad\qquad\qquad (4)$$

**Inductive bias.** We note that Vision Transformer has much less image-specific inductive bias than CNNs. In CNNs, locality, two-dimensional neighborhood structure, and translation equivariance are baked into each layer throughout the whole model. In ViT, only MLP layers are local and translationally equivariant, while the self-attention layers are global. The two-dimensional neighborhood structure is used very sparingly: in the beginning of the model by cutting the image into patches and at fine-tuning time for adjusting the position embeddings for images of different resolution (as described below). Other than that, the position embeddings at initialization time carry no information about the 2D positions of the patches and all spatial relations between the patches have to be learned from scratch.

**Hybrid Architecture.** As an alternative to raw image patches, the input sequence can be formed from feature maps of a CNN (LeCun et al., 1989). In this hybrid model, the patch embedding projection $\mathbf{E}$ (Eq. 1) is applied to patches extracted from a CNN feature map. As a special case, the patches can have spatial size 1x1, which means that the input sequence is obtained by simply flattening the spatial dimensions of the feature map and projecting to the Transformer dimension. The classification input embedding and position embeddings are added as described above.

## 3.2 FINE-TUNING AND HIGHER RESOLUTION

Typically, we pre-train ViT on large datasets, and fine-tune to (smaller) downstream tasks. For this, we remove the pre-trained prediction head and attach a zero-initialized $D \times K$ feedforward layer, where $K$ is the number of downstream classes. It is often beneficial to fine-tune at higher resolution than pre-training (Touvron et al., 2019; Kolesnikov et al., 2020). When feeding images of higher resolution, we keep the patch size the same, which results in a larger effective sequence length. The Vision Transformer can handle arbitrary sequence lengths (up to memory constraints), however, the pre-trained position embeddings may no longer be meaningful. We therefore perform 2D interpolation of the pre-trained position embeddings, according to their location in the original image. Note that this resolution adjustment and patch extraction are the only points at which an inductive bias about the 2D structure of the images is manually injected into the Vision Transformer.

## 4 EXPERIMENTS

We evaluate the representation learning capabilities of ResNet, Vision Transformer (ViT), and the hybrid. To understand the data requirements of each model, we pre-train on datasets of varying size and evaluate many benchmark tasks. When considering the computational cost of pre-training the model, ViT performs very favourably, attaining state of the art on most recognition benchmarks at a lower pre-training cost. Lastly, we perform a small experiment using self-supervision, and show that self-supervised ViT holds promise for the future.

## 4.1 SETUP

**Datasets.** To explore model scalability, we use the ILSVRC-2012 ImageNet dataset with 1k classes and 1.3M images (we refer to it as ImageNet in what follows), its superset ImageNet-21k with 21k classes and 14M images (Deng et al., 2009), and JFT (Sun et al., 2017) with 18k classes and 303M high-resolution images. We de-duplicate the pre-training datasets w.r.t. the test sets of the downstream tasks following Kolesnikov et al. (2020). We transfer the models trained on these dataset to several benchmark tasks: ImageNet on the original validation labels and the cleaned-up ReaL labels (Beyer et al., 2020), CIFAR-10/100 (Krizhevsky, 2009), Oxford-IIIT Pets (Parkhi et al., 2012), and Oxford Flowers-102 (Nilsback & Zisserman, 2008). For these datasets, pre-processing follows Kolesnikov et al. (2020).

| Model | Layers | Hidden size $D$ | MLP size | Heads | Params |
|-------|--------|-----------------|----------|-------|--------|
| ViT-Base | 12 | 768 | 3072 | 12 | 86M |
| ViT-Large | 24 | 1024 | 4096 | 16 | 307M |
| ViT-Huge | 32 | 1280 | 5120 | 16 | 632M |

Table 1: Details of Vision Transformer model variants.

We also evaluate on the 19-task VTAB classification suite (Zhai et al., 2019b). VTAB evaluates low-data transfer to diverse tasks, using 1 000 training examples per task. The tasks are divided into three groups: *Natural* – tasks like the above, Pets, CIFAR, etc. *Specialized* – medical and satellite imagery, and *Structured* – tasks that require geometric understanding like localization.

**Model Variants.** We base ViT configurations on those used for BERT (Devlin et al., 2019), as summarized in Table 1. The "Base" and "Large" models are directly adopted from BERT and we add the larger "Huge" model. In what follows we use brief notation to indicate the model size and the input patch size: for instance, ViT-L/16 means the "Large" variant with $16 \times 16$ input patch size. Note that the Transformer's sequence length is inversely proportional to the square of the patch size, thus models with smaller patch size are computationally more expensive.

For the baseline CNNs, we use ResNet (He et al., 2016), but replace the Batch Normalization layers (Ioffe & Szegedy, 2015) with Group Normalization (Wu & He, 2018), and used standardized convolutions (Qiao et al., 2019). These modifications improve transfer (Kolesnikov et al., 2020), and we denote the modified model "ResNet (BiT)". For the hybrids, we feed the intermediate feature maps into ViT with patch size of one "pixel". To experiment with different sequence lengths, we either (i) take the output of stage 4 of a regular ResNet50 or (ii) remove stage 4, place the same number of layers in stage 3 (keeping the total number of layers), and take the output of this extended stage 3. Option (ii) results in a 4x longer sequence length, and a more expensive ViT model.

**Training & Fine-tuning.** We train all models, including ResNets, using Adam (Kingma & Ba, 2015) with $\beta_1 = 0.9$, $\beta_2 = 0.999$, a batch size of 4096 and apply a high weight decay of $0.1$, which we found to be useful for transfer of all models (Appendix D.1 shows that, in contrast to common practices, Adam works slightly better than SGD for ResNets in our setting). We use a linear learning rate warmup and decay, see Appendix B.1 for details. For fine-tuning we use SGD with momentum, batch size 512, for all models, see Appendix B.1.1. For ImageNet results in Table 2, we fine-tuned at higher resolution: 512 for ViT-L/16 and 518 for ViT-H/14, and also used Polyak & Juditsky (1992) averaging with a factor of 0.9999 (Ramachandran et al., 2019; Wang et al., 2020b).

**Metrics.** We report results on downstream datasets either through few-shot or fine-tuning accuracy. Fine-tuning accuracies capture the performance of each model after fine-tuning it on the respective dataset. Few-shot accuracies are obtained by solving a regularized least-squares regression problem that maps the (frozen) representation of a subset of training images to $\{-1, 1\}^K$ target vectors. This formulation allows us to recover the exact solution in closed form. Though we mainly focus on fine-tuning performance, we sometimes use linear few-shot accuracies for fast on-the-fly evaluation where fine-tuning would be too costly.

## 4.2 Comparison to State of the Art

We first compare our largest models – ViT-H/14 and ViT-L/16 – to state-of-the-art CNNs from the literature. The first comparison point is Big Transfer (BiT) (Kolesnikov et al., 2020), which performs supervised transfer learning with large ResNets. The second is Noisy Student (Xie et al., 2020), which is a large EfficientNet trained using semi-supervised learning on ImageNet and JFT-300M with the labels removed. Currently, Noisy Student is the state of the art on ImageNet and BiT-L on the other datasets reported here. All models were trained on TPUv3 hardware, and we report the number of TPUv3-core-days taken to pre-train each of them, that is, the number of TPU v3 cores (2 per chip) used for training multiplied by the training time in days.

Table 2 shows the results. The smaller ViT-L/16 model pre-trained on JFT-300M outperforms BiT-L (which is pre-trained on the same dataset) on all tasks, while requiring substantially less computational resources to train. The larger model, ViT-H/14, further improves the performance, especially on the more challenging datasets – ImageNet, CIFAR-100, and the VTAB suite. Interestingly, this

|  | Ours-JFT (ViT-H/14) | Ours-JFT (ViT-L/16) | Ours-I21k (ViT-L/16) | BiT-L (ResNet152x4) | Noisy Student (EfficientNet-L2) |
|---|---|---|---|---|---|
| ImageNet | $\mathbf{88.55} \pm {\scriptstyle 0.04}$ | $87.76 \pm {\scriptstyle 0.03}$ | $85.30 \pm {\scriptstyle 0.02}$ | $87.54 \pm {\scriptstyle 0.02}$ | $88.4/88.5^{*}$ |
| ImageNet ReaL | $\mathbf{90.72} \pm {\scriptstyle 0.05}$ | $90.54 \pm {\scriptstyle 0.03}$ | $88.62 \pm {\scriptstyle 0.05}$ | $90.54$ | $90.55$ |
| CIFAR-10 | $\mathbf{99.50} \pm {\scriptstyle 0.06}$ | $99.42 \pm {\scriptstyle 0.03}$ | $99.15 \pm {\scriptstyle 0.03}$ | $99.37 \pm {\scriptstyle 0.06}$ | $-$ |
| CIFAR-100 | $\mathbf{94.55} \pm {\scriptstyle 0.04}$ | $93.90 \pm {\scriptstyle 0.05}$ | $93.25 \pm {\scriptstyle 0.05}$ | $93.51 \pm {\scriptstyle 0.08}$ | $-$ |
| Oxford-IIIT Pets | $\mathbf{97.56} \pm {\scriptstyle 0.03}$ | $97.32 \pm {\scriptstyle 0.11}$ | $94.67 \pm {\scriptstyle 0.15}$ | $96.62 \pm {\scriptstyle 0.23}$ | $-$ |
| Oxford Flowers-102 | $99.68 \pm {\scriptstyle 0.02}$ | $\mathbf{99.74} \pm {\scriptstyle 0.00}$ | $99.61 \pm {\scriptstyle 0.02}$ | $99.63 \pm {\scriptstyle 0.03}$ | $-$ |
| VTAB (19 tasks) | $\mathbf{77.63} \pm {\scriptstyle 0.23}$ | $76.28 \pm {\scriptstyle 0.46}$ | $72.72 \pm {\scriptstyle 0.21}$ | $76.29 \pm {\scriptstyle 1.70}$ | $-$ |
| TPUv3-core-days | 2.5k | 0.68k | 0.23k | 9.9k | 12.3k |

Table 2: Comparison with state of the art on popular image classification benchmarks. We report mean and standard deviation of the accuracies, averaged over three fine-tuning runs. Vision Transformer models pre-trained on the JFT-300M dataset outperform ResNet-based baselines on all datasets, while taking substantially less computational resources to pre-train. ViT pre-trained on the smaller public ImageNet-21k dataset performs well too. *Slightly improved 88.5% result reported in Touvron et al. (2020).

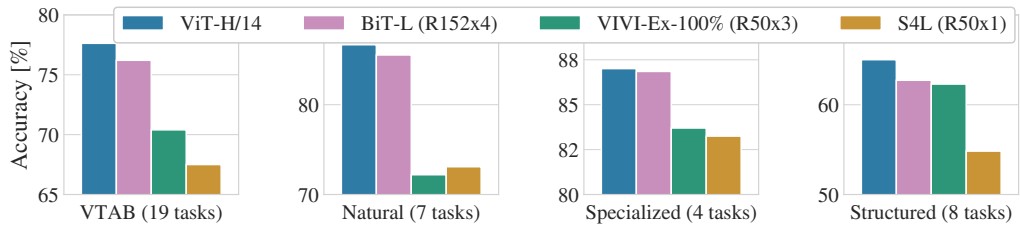

Figure 2: Breakdown of VTAB performance in *Natural*, *Specialized*, and *Structured* task groups.

model still took substantially less compute to pre-train than prior state of the art. However, we note that pre-training efficiency may be affected not only by the architecture choice, but also other parameters, such as training schedule, optimizer, weight decay, etc. We provide a controlled study of performance vs. compute for different architectures in Section 4.4. Finally, the ViT-L/16 model pre-trained on the public ImageNet-21k dataset performs well on most datasets too, while taking fewer resources to pre-train: it could be trained using a standard cloud TPUv3 with 8 cores in approximately 30 days.

Figure 2 decomposes the VTAB tasks into their respective groups, and compares to previous SOTA methods on this benchmark: BiT, VIVI – a ResNet co-trained on ImageNet and Youtube (Tschannen et al., 2020), and S4L – supervised plus semi-supervised learning on ImageNet (Zhai et al., 2019a). ViT-H/14 outperforms BiT-R152x4, and other methods, on the *Natural* and *Structured* tasks. On the *Specialized* the performance of the top two models is similar.

### 4.3 PRE-TRAINING DATA REQUIREMENTS

The Vision Transformer performs well when pre-trained on a large JFT-300M dataset. With fewer inductive biases for vision than ResNets, how crucial is the dataset size? We perform two series of experiments.

First, we pre-train ViT models on datasets of increasing size: ImageNet, ImageNet-21k, and JFT-300M. To boost the performance on the smaller datasets, we optimize three basic regularization parameters – weight decay, dropout, and label smoothing. Figure 3 shows the results after fine-tuning to ImageNet (results on other datasets are shown in Table 5)[2]. When pre-trained on the smallest dataset, ImageNet, ViT-Large models underperform compared to ViT-Base models, despite (moderate) regularization. With ImageNet-21k pre-training, their performances are similar. Only with JFT-300M, do we see the full benefit of larger models. Figure 3 also shows the performance

---

[2]Note that the ImageNet pre-trained models are also fine-tuned, but again on ImageNet. This is because the resolution increase during fine-tuning improves the performance.

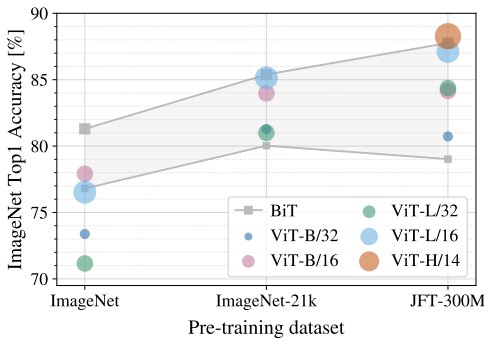 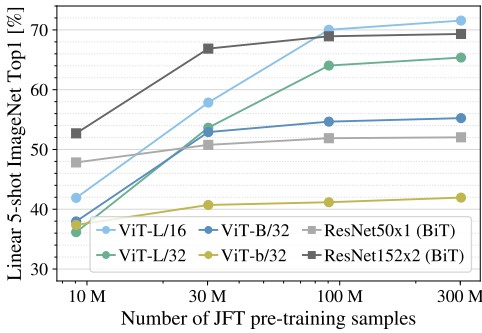

Figure 3: Transfer to ImageNet. While large ViT models perform worse than BiT ResNets (shaded area) when pre-trained on small datasets, they shine when pre-trained on larger datasets. Similarly, larger ViT variants overtake smaller ones as the dataset grows.

Figure 4: Linear few-shot evaluation on ImageNet versus pre-training size. ResNets perform better with smaller pre-training datasets but plateau sooner than ViT, which performs better with larger pre-training. ViT-b is ViT-B with all hidden dimensions halved.

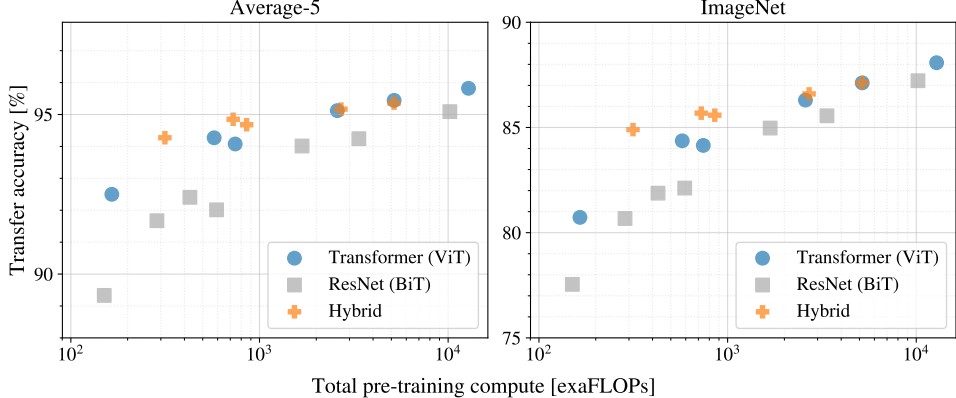

Figure 5: Performance versus cost for different architectures: Vision Transformers, ResNets, and hybrids. Vision Transformers generally outperform ResNets with the same computational budget. Hybrids improve upon pure Transformers for smaller model sizes, but the gap vanishes for larger models.

region spanned by BiT models of different sizes. The BiT CNNs outperform ViT on ImageNet, but with the larger datasets, ViT overtakes.

Second, we train our models on random subsets of 9M, 30M, and 90M as well as the full JFT-300M dataset. We do not perform additional regularization on the smaller subsets and use the same hyper-parameters for all settings. This way, we assess the intrinsic model properties, and not the effect of regularization. We do, however, use early-stopping, and report the best validation accuracy achieved during training. To save compute, we report few-shot linear accuracy instead of full fine-tuning accuracy. Figure 4 contains the results. Vision Transformers overfit more than ResNets with comparable computational cost on smaller datasets. For example, ViT-B/32 is slightly faster than ResNet50; it performs much worse on the 9M subset, but better on 90M+ subsets. The same is true for ResNet152x2 and ViT-L/16. This result reinforces the intuition that the convolutional inductive bias is useful for smaller datasets, but for larger ones, learning the relevant patterns directly from data is sufficient, even beneficial.

Overall, the few-shot results on ImageNet (Figure 4), as well as the low-data results on VTAB (Table 2) seem promising for very low-data transfer. Further analysis of few-shot properties of ViT is an exciting direction of future work.

## 4.4 SCALING STUDY

We perform a controlled scaling study of different models by evaluating transfer performance from JFT-300M. In this setting data size does not bottleneck the models' performances, and we assess performance versus pre-training cost of each model. The model set includes: 7 ResNets, R50x1, R50x2 R101x1, R152x1, R152x2, pre-trained for 7 epochs, plus R152x2 and R200x3 pre-trained for 14 epochs; 6 Vision Transformers, ViT-B/32, B/16, L/32, L/16, pre-trained for 7 epochs, plus L/16 and H/14 pre-trained for 14 epochs; and 5 hybrids, R50+ViT-B/32, B/16, L/32, L/16 pre-trained for 7 epochs, plus R50+ViT-L/16 pre-trained for 14 epochs (for hybrids, the number at the end of the model name stands not for the patch size, but for the total dowsampling ratio in the ResNet backbone).

Figure 5 contains the transfer performance versus total pre-training compute (see Appendix D.4 for details on computational costs). Detailed results per model are provided in Table 6 in the Appendix. A few patterns can be observed. First, Vision Transformers dominate ResNets on the performance/compute trade-off. ViT uses approximately $2 - 4\times$ less compute to attain the same performance (average over 5 datasets). Second, hybrids slightly outperform ViT at small computational budgets, but the difference vanishes for larger models. This result is somewhat surprising, since one might expect convolutional local feature processing to assist ViT at any size. Third, Vision Transformers appear not to saturate within the range tried, motivating future scaling efforts.

## 4.5 INSPECTING VISION TRANSFORMER

To begin to understand how the Vision Transformer processes image data, we analyze its internal representations. The first layer of the Vision Transformer linearly projects the flattened patches into a lower-dimensional space (Eq. 1). Figure 7 (left) shows the top principal components of the the learned embedding filters. The components resemble plausible basis functions for a low-dimensional representation of the fine structure within each patch.

After the projection, a learned position embedding is added to the patch representations. Figure 7 (center) shows that the model learns to encode distance within the image in the similarity of position embeddings, i.e. closer patches tend to have more similar position embeddings. Further, the row-column structure appears; patches in the same row/column have similar embeddings. Finally, a sinusoidal structure is sometimes apparent for larger grids (Appendix D). That the position embeddings learn to represent 2D image topology explains why hand-crafted 2D-aware embedding variants do not yield improvements (Appendix D.3).

Self-attention allows ViT to integrate information across the entire image even in the lowest layers. We investigate to what degree the network makes use of this capability. Specifically, we compute the average distance in image space across which information is integrated, based on the attention weights (Figure 7, right). This "attention distance" is analogous to receptive field size in CNNs.

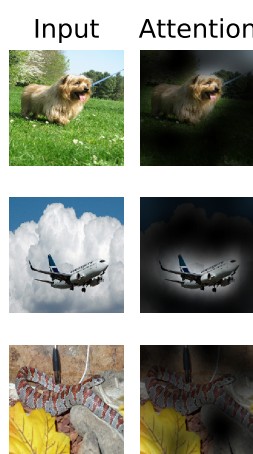

Input    Attention

Figure 6: Representative examples of attention from the output token to the input space. See Appendix D.6 for details.

We find that some heads attend to most of the image already in the lowest layers, showing that the ability to integrate information globally is indeed used by the model. Other attention heads have consistently small attention distances in the low layers. This highly localized attention is less pronounced in hybrid models that apply a ResNet before the Transformer (Figure 7, right), suggesting that it may serve a similar function as early convolutional layers in CNNs. Further, the attention distance increases with network depth. Globally, we find that the model attends to image regions that are semantically relevant for classification (Figure 6).

## 4.6 SELF-SUPERVISION

Transformers show impressive performance on NLP tasks. However, much of their success stems not only from their excellent scalability but also from large scale self-supervised pre-training (Devlin

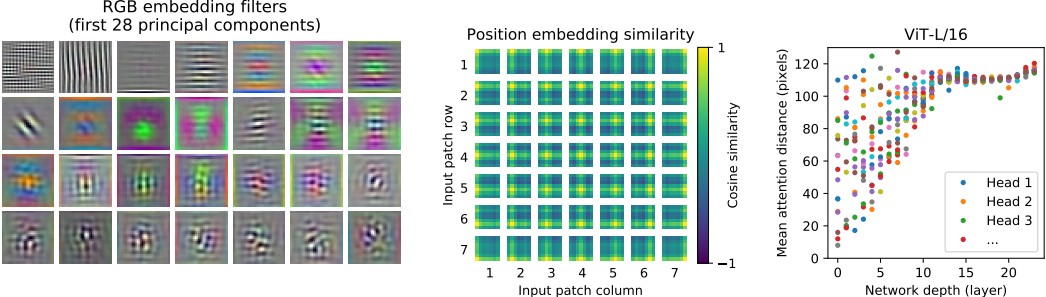

Figure 7: **Left:** Filters of the initial linear embedding of RGB values of ViT-L/32. **Center:** Similarity of position embeddings of ViT-L/32. Tiles show the cosine similarity between the position embedding of the patch with the indicated row and column and the position embeddings of all other patches. **Right:** Size of attended area by head and network depth. Each dot shows the mean attention distance across images for one of 16 heads at one layer. See Appendix D.6 for details.

et al., 2019; Radford et al., 2018). We also perform a preliminary exploration on *masked patch prediction* for self-supervision, mimicking the masked language modeling task used in BERT. With self-supervised pre-training, our smaller ViT-B/16 model achieves 79.9% accuracy on ImageNet, a significant improvement of 2% to training from scratch, but still 4% behind supervised pre-training. Appendix B.1.2 contains further details. We leave exploration of contrastive pre-training (Chen et al., 2020b; He et al., 2020; Bachman et al., 2019; Hénaff et al., 2020) to future work.

## 5 CONCLUSION

We have explored the direct application of Transformers to image recognition. Unlike prior works using self-attention in computer vision, we do not introduce image-specific inductive biases into the architecture apart from the initial patch extraction step. Instead, we interpret an image as a sequence of patches and process it by a standard Transformer encoder as used in NLP. This simple, yet scalable, strategy works surprisingly well when coupled with pre-training on large datasets. Thus, Vision Transformer matches or exceeds the state of the art on many image classification datasets, whilst being relatively cheap to pre-train.

While these initial results are encouraging, many challenges remain. One is to apply ViT to other computer vision tasks, such as detection and segmentation. Our results, coupled with those in Carion et al. (2020), indicate the promise of this approach. Another challenge is to continue exploring self-supervised pre-training methods. Our initial experiments show improvement from self-supervised pre-training, but there is still large gap between self-supervised and large-scale supervised pre-training. Finally, further scaling of ViT would likely lead to improved performance.

## ACKNOWLEDGEMENTS

The work was performed in Berlin, Zürich, and Amsterdam. We thank many colleagues at Google for their help, in particular Andreas Steiner for crucial help with the infrastructure and the open-source release of the code; Joan Puigcerver and Maxim Neumann for help with the large-scale training infrastructure; Dmitry Lepikhin, Aravindh Mahendran, Daniel Keysers, Mario Lučić, Noam Shazeer, and Colin Raffel for useful discussions.

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

| Models | Dataset | Epochs | Base LR | LR decay | Weight decay | Dropout |
|---|---|---|---|---|---|---|
| ViT-B/{16,32} | JFT-300M | 7 | $8 \cdot 10^{-4}$ | linear | 0.1 | 0.0 |
| ViT-L/32 | JFT-300M | 7 | $6 \cdot 10^{-4}$ | linear | 0.1 | 0.0 |
| ViT-L/16 | JFT-300M | 7/14 | $4 \cdot 10^{-4}$ | linear | 0.1 | 0.0 |
| ViT-H/14 | JFT-300M | 14 | $3 \cdot 10^{-4}$ | linear | 0.1 | 0.0 |
| R50x{1,2} | JFT-300M | 7 | $10^{-3}$ | linear | 0.1 | 0.0 |
| R101x1 | JFT-300M | 7 | $8 \cdot 10^{-4}$ | linear | 0.1 | 0.0 |
| R152x{1,2} | JFT-300M | 7 | $6 \cdot 10^{-4}$ | linear | 0.1 | 0.0 |
| R50+ViT-B/{16,32} | JFT-300M | 7 | $8 \cdot 10^{-4}$ | linear | 0.1 | 0.0 |
| R50+ViT-L/32 | JFT-300M | 7 | $2 \cdot 10^{-4}$ | linear | 0.1 | 0.0 |
| R50+ViT-L/16 | JFT-300M | 7/14 | $4 \cdot 10^{-4}$ | linear | 0.1 | 0.0 |
| ViT-B/{16,32} | ImageNet-21k | 90 | $10^{-3}$ | linear | 0.03 | 0.1 |
| ViT-L/{16,32} | ImageNet-21k | 30/90 | $10^{-3}$ | linear | 0.03 | 0.1 |
| ViT-* | ImageNet | 300 | $3 \cdot 10^{-3}$ | cosine | 0.3 | 0.1 |

Table 3: Hyperparameters for training. All models are trained with a batch size of 4096 and learning rate warmup of 10k steps. For ImageNet we found it beneficial to additionally apply gradient clipping at global norm 1. Training resolution is 224.

## APPENDIX

## A MULTIHEAD SELF-ATTENTION

Standard **qkv** self-attention (SA, Vaswani et al. (2017)) is a popular building block for neural architectures. For each element in an input sequence $\mathbf{z} \in \mathbb{R}^{N \times D}$, we compute a weighted sum over all values $\mathbf{v}$ in the sequence. The attention weights $A_{ij}$ are based on the pairwise similarity between two elements of the sequence and their respective query $\mathbf{q}^i$ and key $\mathbf{k}^j$ representations.

$$[\mathbf{q}, \mathbf{k}, \mathbf{v}] = \mathbf{z}\mathbf{U}_{qkv} \qquad\qquad \mathbf{U}_{qkv} \in \mathbb{R}^{D \times 3D_h}, \qquad (5)$$

$$A = \mathrm{softmax}\left(\mathbf{q}\mathbf{k}^\top / \sqrt{D_h}\right) \qquad\qquad A \in \mathbb{R}^{N \times N}, \qquad (6)$$

$$\mathrm{SA}(\mathbf{z}) = A\mathbf{v}. \qquad (7)$$

Multihead self-attention (MSA) is an extension of SA in which we run $k$ self-attention operations, called "heads", in parallel, and project their concatenated outputs. To keep compute and number of parameters constant when changing $k$, $D_h$ (Eq. 5) is typically set to $D/k$.

$$\mathrm{MSA}(\mathbf{z}) = [\mathrm{SA}_1(z); \mathrm{SA}_2(z); \cdots ; \mathrm{SA}_k(z)]\, \mathbf{U}_{msa} \qquad \mathbf{U}_{msa} \in \mathbb{R}^{k \cdot D_h \times D} \qquad (8)$$

## B EXPERIMENT DETAILS

### B.1 TRAINING

Table 3 summarizes our training setups for our different models. We found strong regularization to be key when training models from scratch on ImageNet. Dropout, when used, is applied after every dense layer except for the the qkv-projections and directly after adding positional- to patch embeddings. Hybrid models are trained with the exact setup as their ViT counterparts. Finally, all training is done on resolution 224.

### B.1.1 FINE-TUNING

We fine-tune all ViT models using SGD with a momentum of 0.9. We run a small grid search over learning rates, see learning rate ranges in Table 4. To do so, we use small sub-splits from the training set (10% for Pets and Flowers, 2% for CIFAR, 1% ImageNet) as development set and train on the remaining data. For final results we train on the entire training set and evaluate on the respective test data. For fine-tuning ResNets and hybrid models we use the exact same setup, with the only exception of ImageNet where we add another value 0.06 to the learning rate sweep. Additionally,

| Dataset | Steps | Base LR |
|---|---|---|
| ImageNet | 20 000 | {0.003, 0.01, 0.03, 0.06} |
| CIFAR100 | 10 000 | {0.001, 0.003, 0.01, 0.03} |
| CIFAR10 | 10 000 | {0.001, 0.003, 0.01, 0.03} |
| Oxford-IIIT Pets | 500 | {0.001, 0.003, 0.01, 0.03} |
| Oxford Flowers-102 | 500 | {0.001, 0.003, 0.01, 0.03} |
| VTAB (19 tasks) | 2 500 | 0.01 |

Table 4: Hyperparameters for fine-tuning. All models are fine-tuned with cosine learning rate decay, a batch size of 512, no weight decay, and grad clipping at global norm 1. If not mentioned otherwise, fine-tuning resolution is 384.

for ResNets we also run the setup of Kolesnikov et al. (2020) and select the best results across this run and our sweep. Finally, if not mentioned otherwise, all fine-tuning experiments run at 384 resolution (running fine-tuning at different resolution than training is common practice (Kolesnikov et al., 2020)).

When transferring ViT models to another dataset, we remove the whole head (two linear layers) and replace it by a single, zero-initialized linear layer outputting the number of classes required by the target dataset. We found this to be a little more robust than simply re-initializing the very last layer.

For VTAB we follow the protocol in Kolesnikov et al. (2020), and use the same hyperparameter setting for all tasks. We use a learning rate of $0.01$ and train for 2500 steps (Tab. 4). We chose this setting by running a small sweep over two learning rates and two schedules, and selecting the setting with the highest VTAB score on the 200-example validation sets. We follow the pre-processing used in Kolesnikov et al. (2020), except that we do not use task-specific input resolutions. Instead we find that Vision Transformer benefits most from a high resolution ($384 \times 384$) for all tasks.

### B.1.2 SELF-SUPERVISION

We employ the *masked patch prediction* objective for preliminary self-supervision experiments. To do so we corrupt 50% of patch embeddings by either replacing their embeddings with a learnable [mask] embedding (80%), a random other patch embedding (10%) or just keeping them as is (10%). This setup is very similar to the one used for language by Devlin et al. (2019). Finally, we predict the 3-bit, mean color (i.e., 512 colors in total) of every corrupted patch using their respective patch representations.

We trained our self-supervised model for 1M steps (ca. 14 epochs) with batch size 4096 on JFT. We use Adam, with a base learning rate of $2 \cdot 10^{-4}$, warmup of 10k steps and cosine learning rate decay. As prediction targets for pretraining we tried the following settings: 1) predicting only the mean, 3bit color (i.e., 1 prediction of 512 colors), 2) predicting a $4 \times 4$ downsized version of the $16 \times 16$ patch with 3bit colors in parallel (i.e., 16 predictions of 512 colors), 3) regression on the full patch using L2 (i.e., 256 regressions on the 3 RGB channels). Surprisingly, we found that all worked quite well, though L2 was slightly worse. We report final results only for option 1) because it has shown best few-shot performance. We also experimented with 15% corruption rate as used by Devlin et al. (2019) but results were also slightly worse on our few-shot metrics.

Lastly, we would like to remark that our instantiation of masked patch prediction doesn't require such an enormous amount of pretraining nor a large dataset such as JFT in order to lead to similar performance gains on ImageNet classification. That is, we observed diminishing returns on downstream performance after 100k pretraining steps, and see similar gains when pretraining on ImageNet.

## C ADDITIONAL RESULTS

We report detailed results corresponding to the figures presented in the paper. Table 5 corresponds to Figure 3 from the paper and shows transfer performance of different ViT models pre-trained on datasets of increasing size: ImageNet, ImageNet-21k, and JFT-300M. Table 6 corresponds to

|  |  | ViT-B/16 | ViT-B/32 | ViT-L/16 | ViT-L/32 | ViT-H/14 |
|---|---|---|---|---|---|---|
| ImageNet | CIFAR-10 | 98.13 | 97.77 | 97.86 | 97.94 | - |
|  | CIFAR-100 | 87.13 | 86.31 | 86.35 | 87.07 | - |
|  | ImageNet | 77.91 | 73.38 | 76.53 | 71.16 | - |
|  | ImageNet ReaL | 83.57 | 79.56 | 82.19 | 77.83 | - |
|  | Oxford Flowers-102 | 89.49 | 85.43 | 89.66 | 86.36 | - |
|  | Oxford-IIIT-Pets | 93.81 | 92.04 | 93.64 | 91.35 | - |
| ImageNet-21k | CIFAR-10 | 98.95 | 98.79 | 99.16 | 99.13 | 99.27 |
|  | CIFAR-100 | 91.67 | 91.97 | 93.44 | 93.04 | 93.82 |
|  | ImageNet | 83.97 | 81.28 | 85.15 | 80.99 | 85.13 |
|  | ImageNet ReaL | 88.35 | 86.63 | 88.40 | 85.65 | 88.70 |
|  | Oxford Flowers-102 | 99.38 | 99.11 | 99.61 | 99.19 | 99.51 |
|  | Oxford-IIIT-Pets | 94.43 | 93.02 | 94.73 | 93.09 | 94.82 |
| JFT-300M | CIFAR-10 | 99.00 | 98.61 | 99.38 | 99.19 | 99.50 |
|  | CIFAR-100 | 91.87 | 90.49 | 94.04 | 92.52 | 94.55 |
|  | ImageNet | 84.15 | 80.73 | 87.12 | 84.37 | 88.04 |
|  | ImageNet ReaL | 88.85 | 86.27 | 89.99 | 88.28 | 90.33 |
|  | Oxford Flowers-102 | 99.56 | 99.27 | 99.56 | 99.45 | 99.68 |
|  | Oxford-IIIT-Pets | 95.80 | 93.40 | 97.11 | 95.83 | 97.56 |

Table 5: Top1 accuracy (in %) of Vision Transformer on various datasets when pre-trained on ImageNet, ImageNet-21k or JFT300M. These values correspond to Figure 3 in the main text. Models are fine-tuned at 384 resolution. Note that the ImageNet results are computed without additional techniques (Polyak averaging and 512 resolution images) used to achieve results in Table 2.

| Model | Epochs | ImageNet | ImageNet ReaL | CIFAR-10 | CIFAR-100 | Pets | Flowers | exaFLOPs |
|---|---|---|---|---|---|---|---|---|
| ViT-B/32 | 7 | 80.73 | 86.27 | 98.61 | 90.49 | 93.40 | 99.27 | 164 |
| ViT-B/16 | 7 | 84.15 | 88.85 | 99.00 | 91.87 | 95.80 | 99.56 | 743 |
| ViT-L/32 | 7 | 84.37 | 88.28 | 99.19 | 92.52 | 95.83 | 99.45 | 574 |
| ViT-L/16 | 7 | 86.30 | 89.43 | 99.38 | 93.46 | 96.81 | 99.66 | 2586 |
| ViT-L/16 | 14 | 87.12 | 89.99 | 99.38 | 94.04 | 97.11 | 99.56 | 5172 |
| ViT-H/14 | 14 | 88.08 | 90.36 | 99.50 | 94.71 | 97.11 | 99.71 | 12826 |
| ResNet50x1 | 7 | 77.54 | 84.56 | 97.67 | 86.07 | 91.11 | 94.26 | 150 |
| ResNet50x2 | 7 | 82.12 | 87.94 | 98.29 | 89.20 | 93.43 | 97.02 | 592 |
| ResNet101x1 | 7 | 80.67 | 87.07 | 98.48 | 89.17 | 94.08 | 95.95 | 285 |
| ResNet152x1 | 7 | 81.88 | 87.96 | 98.82 | 90.22 | 94.17 | 96.94 | 427 |
| ResNet152x2 | 7 | 84.97 | 89.69 | 99.06 | 92.05 | 95.37 | 98.62 | 1681 |
| ResNet152x2 | 14 | 85.56 | 89.89 | 99.24 | 91.92 | 95.75 | 98.75 | 3362 |
| ResNet200x3 | 14 | 87.22 | 90.15 | 99.34 | 93.53 | 96.32 | 99.04 | 10212 |
| R50x1+ViT-B/32 | 7 | 84.90 | 89.15 | 99.01 | 92.24 | 95.75 | 99.46 | 315 |
| R50x1+ViT-B/16 | 7 | 85.58 | 89.65 | 99.14 | 92.63 | 96.65 | 99.40 | 855 |
| R50x1+ViT-L/32 | 7 | 85.68 | 89.04 | 99.24 | 92.93 | 96.97 | 99.43 | 725 |
| R50x1+ViT-L/16 | 7 | 86.60 | 89.72 | 99.18 | 93.64 | 97.03 | 99.40 | 2704 |
| R50x1+ViT-L/16 | 14 | 87.12 | 89.76 | 99.31 | 93.89 | 97.36 | 99.11 | 5165 |

Table 6: Detailed results of model scaling experiments. These correspond to Figure 5 in the main paper.

Figure 5 from the paper and shows the transfer performance of ViT, ResNet, and hybrid models of varying size, as well as the estimated computational cost of their pre-training.

# D  ADDITIONAL ANALYSES

## D.1  SGD VS. ADAM FOR RESNETS

ResNets are typically trained with SGD and our use of Adam as optimizer is quite unconventional. Here we show the experiments that motivated this choice. Namely, we compare the fine-tuning performance of two ResNets – 50x1 and 152x2 – pre-trained on JFT with SGD and Adam. For SGD, we use the hyperparameters recommended by Kolesnikov et al. (2020). Results are presented

| Dataset | ResNet50 | | ResNet152x2 | |
|---|---|---|---|---|
| | Adam | SGD | Adam | SGD |
| ImageNet | 77.54 | 78.24 | 84.97 | 84.37 |
| CIFAR10 | 97.67 | 97.46 | 99.06 | 99.07 |
| CIFAR100 | 86.07 | 85.17 | 92.05 | 91.06 |
| Oxford-IIIT Pets | 91.11 | 91.00 | 95.37 | 94.79 |
| Oxford Flowers-102 | 94.26 | 92.06 | 98.62 | 99.32 |
| Average | 89.33 | 88.79 | 94.01 | 93.72 |

Table 7: Fine-tuning ResNet models pre-trained with Adam and SGD.

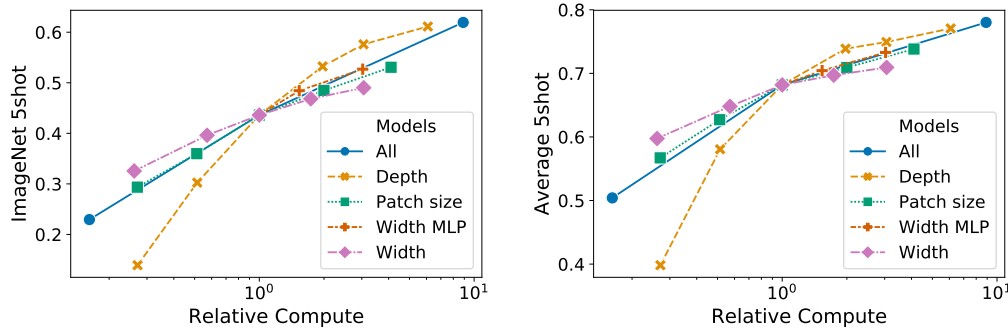

Figure 8: Scaling different model dimensions of the Vision Transformer.

in Table 7. Adam pre-training outperforms SGD pre-training on most datasets and on average. This justifies the choice of Adam as the optimizer used to pre-train ResNets on JFT. Note that the absolute numbers are lower than those reported by Kolesnikov et al. (2020), since we pre-train only for 7 epochs, not 30.

## D.2 TRANSFORMER SHAPE

We ran ablations on scaling different dimensions of the Transformer architecture to find out which are best suited for scaling to very large models. Figure 8 shows 5-shot performance on ImageNet for different configurations. All configurations are based on a ViT model with 8 layers, $D = 1024$, $D_{MLP} = 2048$ and a patch size of 32, the intersection of all lines. We can see that scaling the depth results in the biggest improvements which are clearly visible up until 64 layers. However, diminishing returns are already visible after 16 layers. Interestingly, scaling the width of the network seems to result in the smallest changes. Decreasing the patch size and thus increasing the effective sequence length shows surprisingly robust improvements without introducing parameters. These findings suggest that compute might be a better predictor of performance than the number of parameters, and that scaling should emphasize depth over width if any. Overall, we find that scaling all dimensions proportionally results in robust improvements.

## D.3 POSITIONAL EMBEDDING

We ran ablations on different ways of encoding spatial information using positional embedding. We tried the following cases:

- Providing no positional information: Considering the inputs as a *bag of patches*.

- 1-dimensional positional embedding: Considering the inputs as a sequence of patches in the raster order (default across all other experiments in this paper).

- 2-dimensional positional embedding: Considering the inputs as a grid of patches in two dimensions. In this case, two sets of embeddings are learned, each for one of the axes, $X$-embedding, and $Y$-embedding, each with size $D/2$. Then, based on the coordinate on

| Pos. Emb. | Default/Stem | Every Layer | Every Layer-Shared |
|---|---|---|---|
| No Pos. Emb. | 0.61382 | N/A | N/A |
| 1-D Pos. Emb. | 0.64206 | 0.63964 | 0.64292 |
| 2-D Pos. Emb. | 0.64001 | 0.64046 | 0.64022 |
| Rel. Pos. Emb. | 0.64032 | N/A | N/A |

Table 8: Results of the ablation study on positional embeddings with ViT-B/16 model evaluated on ImageNet 5-shot linear.

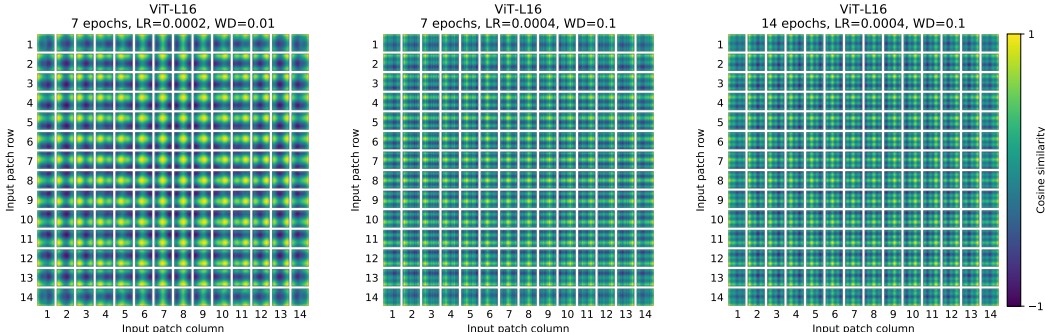

Figure 9: Position embeddings of models trained with different hyperparameters.

the path in the input, we concatenate the $X$ and $Y$ embedding to get the final positional embedding for that patch.

- Relative positional embeddings: Considering the relative distance between patches to encode the spatial information as instead of their absolute position. To do so, we use 1-dimensional Relative Attention, in which we define the relative distance all possible pairs of patches. Thus, for every given pair (one as query, and the other as key/value in the attention mechanism), we have an offset $p_q - p_k$, where each offset is associated with an embedding. Then, we simply run extra attention, where we use the original query (the content of query), but use relative positional embeddings as keys. We then use the logits from the relative attention as a bias term and add it to the logits of the main attention (content-based attention) before applying the softmax.

In addition to different ways of encoding spatial information, we also tried different ways of incorporating this information in our model. For the 1-dimensional and 2-dimensional positional embeddings, we tried three different cases: (1) add positional embeddings to the inputs right after the stem of them model and before feeding the inputs to the Transformer encoder (default across all other experiments in this paper); (2) learn and add positional embeddings to the inputs at the beginning of each layer; (3) add a learned positional embeddings to the inputs at the beginning of each layer (shared between layers).

Table 8 summarizes the results from this ablation study on a ViT-B/16 model. As we can see, while there is a large gap between the performances of the model with no positional embedding and models with positional embedding, there is little to no difference between different ways of encoding positional information. We speculate that since our Transformer encoder operates on patch-level inputs, as opposed to pixel-level, the differences in how to encode spatial information is less important. More precisely, in patch-level inputs, the spatial dimensions are much smaller than the original pixel-level inputs, e.g., $14 \times 14$ as opposed to $224 \times 224$, and learning to represent the spatial relations in this resolution is equally easy for these different positional encoding strategies. Even so, the specific pattern of position embedding similarity learned by the network depends on the training hyperparameters (Figure 9).

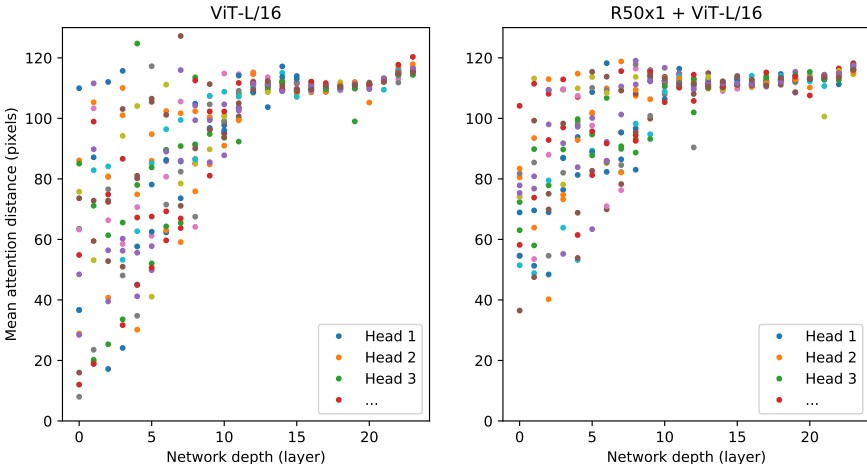

Figure 10: Size of attended area by head and network depth. Attention distance was computed for 128 example images by averaging the distance between the query pixel and all other pixels, weighted by the attention weight. Each dot shows the mean attention distance across images for one of 16 heads at one layer. Image width is 224 pixels.

## D.4 EMPIRICAL COMPUTATIONAL COSTS

We are also interested in real-world speed of the architectures on our hardware, which is not always well predicted by theoretical FLOPs due to details like lane widths and cache sizes. For this purpose, we perform timing of inference speed for the main models of interest, on a TPUv3 accelerator; the difference between inference and backprop speed is a constant model-independent factor.

Figure 11 (left) shows how many images one core can handle per second, across various input sizes. Every single point refers to the peak performance measured across a wide range of batch-sizes. As can be seen, the theoretical bi-quadratic scaling of ViT with image size only barely starts happening for the largest models at the largest resolutions.

Another quantity of interest is the largest batch-size each model can fit onto a core, larger being better for scaling to large datasets. Figure 11 (right) shows this quantity for the same set of models. This shows that large ViT models have a clear advantage in terms of memory-efficiency over ResNet models.

## D.5 AXIAL ATTENTION

Axial Attention (Huang et al., 2020; Ho et al., 2019) is a simple, yet effective technique to run self-attention on large inputs that are organized as multidimensional tensors. The general idea of axial attention is to perform multiple attention operations, each along a single axis of the input tensor, instead of applying 1-dimensional attention to the flattened version of the input. In axial attention, each attention mixes information along a particular axis, while keeping information along the other axes independent. Along this line, Wang et al. (2020b) proposed the AxialResNet model in which all the convolutions with kernel size $3 \times 3$ in a ResNet50 are replaced by axial self-attention, i.e. a row and column attention, augmented by relative positional encoding. We have implemented AxialResNet as a baseline model.[3].

Moreover, we have modified ViT to process inputs in the 2-dimensional shape, instead of a 1-dimensional sequence of patches, and incorporate Axial Transformer blocks, in which instead of

---

[3]Our implementation is based on the open-sourced PyTorch implementation in `https://github.com/csrhddlam/axial-deeplab`. In our experiments, we reproduced the scores reported in (Wang et al., 2020b) in terms of accuracy, however, our implementation, similar to the open-source implementation, is very slow on TPUs. Therefore, we were not able to use it for extensive large-scale experiments. These may be unlocked by a carefully optimized implementation.

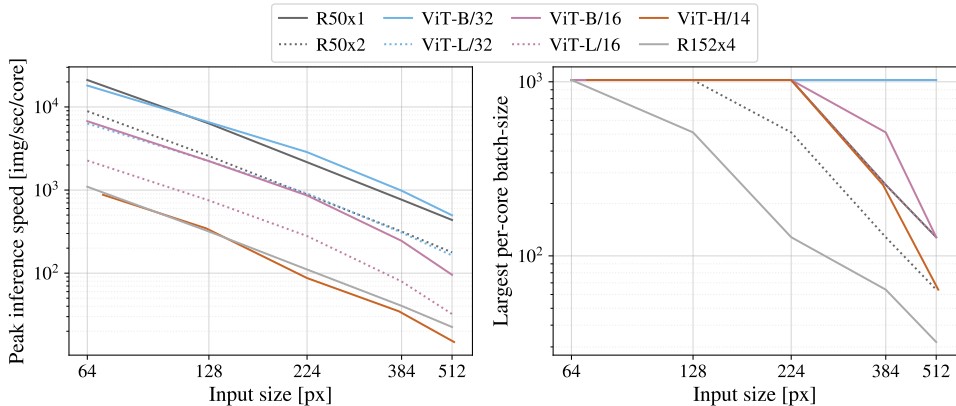

Figure 11: **Left:** Real wall-clock timings of various architectures across input sizes. ViT models have speed comparable to similar ResNets. **Right**: Largest per-core batch-size fitting on device with various architectures across input sizes. ViT models are clearly more memory-efficient.

a self-attention followed by an MLP, we have a a row-self-attention plus an MLP followed by a column-self-attention plus an MLP.

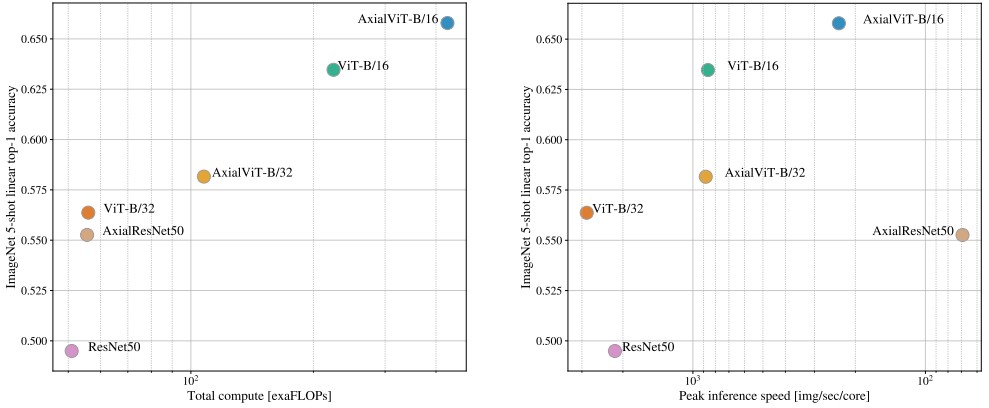

Figure 12: Performance of Axial-Attention based models, in terms of top-1 accuracy on ImageNet 5-shot linear, versus their speed in terms of number of FLOPs (**left**) and inference time (**left**).

Figure 12, present the performance of Axial ResNet, Axial-ViT-B/32 and Axial-ViT-B/16 on ImageNet 5shot linear, when pretrained on JFT dataset, verses the pretraining compute, both in terms of number of FLOPs and inference time (example per seconds). As we can see, both Axial-ViT-B/32 and Axial-ViT-B/16 do better than their ViT-B counterpart in terms of performance, but it comes at the cost of more compute. This is because in Axial-ViT models, each Transformer block with global self-attention is replaced by two Axial Transformer blocks, one with row and one with column self-attention and although the sequence length that self-attention operates on is smaller in axial case, there is a extra MLP per Axial-ViT block. For the AxialResNet, although it looks reasonable in terms of accuracy/compute trade-off (Figure 12, left), the naive implementation is extremely slow on TPUs (Figure 12, right).

### D.6   ATTENTION DISTANCE

To understand how ViT uses self-attention to integrate information across the image, we analyzed the average distance spanned by attention weights at different layers (Figure 10). This "attention distance" is analogous to receptive field size in CNNs. Average attention distance is highly variable

across heads in lower layers, with some heads attending to much of the image, while others attend to small regions at or near the query location. As depth increases, attention distance increases for all heads. In the second half of the network, most heads attend widely across tokens.

## D.7 ATTENTION MAPS

To compute maps of the attention from the output token to the input space (Figures 6 and 13), we used Attention Rollout (Abnar & Zuidema, 2020). Briefly, we averaged attention weights of ViT-L/16 across all heads and then recursively multiplied the weight matrices of all layers. This accounts for the mixing of attention across tokens through all layers.

## D.8 VTAB BREAKDOWN

Table 9 shows the scores attained on each of the VTAB-1k tasks.

Table 9: Breakdown of VTAB-1k performance across tasks.

| | Caltech101 | CIFAR-100 | DTD | Flowers102 | Pets | Sun397 | SVHN | Camelyon | EuroSAT | Resisc45 | Retinopathy | Clevr-Count | Clevr-Dist | DMLab | dSpr-Loc | dSpr-Ori | KITTI-Dist | sNORB-Azim | sNORB-Elev | Mean |
|---|---|---|---|---|---|---|---|---|---|---|---|---|---|---|---|---|---|---|---|---|
| ViT-H/14 (JFT) | 95.3 | 85.5 | 75.2 | 99.7 | 97.2 | 65.0 | 88.9 | 83.3 | 96.7 | 91.4 | 76.6 | 91.7 | 63.8 | 53.1 | 79.4 | 63.3 | 84.5 | 33.2 | 51.2 | 77.6 |
| ViT-L/16 (JFT) | 95.4 | 81.9 | 74.3 | 99.7 | 96.7 | 63.5 | 87.4 | 83.6 | 96.5 | 89.7 | 77.1 | 86.4 | 63.1 | 49.7 | 74.5 | 60.5 | 82.2 | 36.2 | 51.1 | 76.3 |
| ViT-L/16 (I21k) | 90.8 | 84.1 | 74.1 | 99.3 | 92.7 | 61.0 | 80.9 | 82.5 | 95.6 | 85.2 | 75.3 | 70.3 | 56.1 | 41.9 | 74.7 | 64.9 | 79.9 | 30.5 | 41.7 | 72.7 |

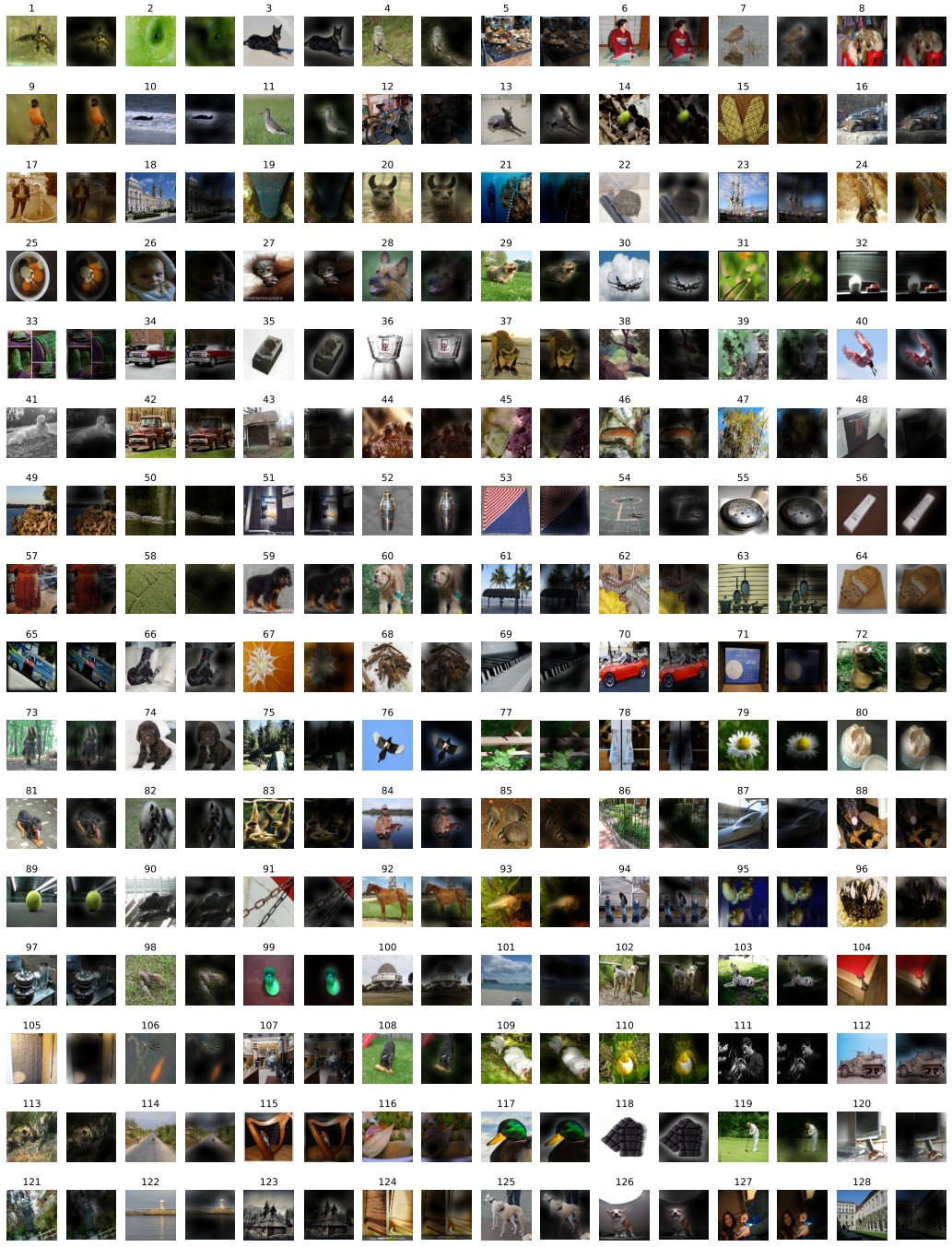

Figure 13: Further example attention maps as in Figure 6 (random selection).

