# OpenReview forum: "An Image is Worth 16x16 Words: Transformers for Image Recognition at Scale"
_ICLR.cc/2021/Conference — ICLR 2021 Oral_

### Official Review · AnonReviewer4 · 2020-10-27
**Transformers cannot yet be a competitor for CNN in different computer vision tasks**

**Rating:** 7
**Confidence:** 3

**Review:**

Following the transformer successes in NLP, this paper explores the performance of a vanilla Transformer (with few simple modifications) for the task of image classification.  It has been experimentally validated that this Transformer (ViT) can attain superior performance for this task compared to SOTA CNN architectures if it is pre-trained on large amounts of data and then applied to mid-sized or small image classification benchmarks. Moreover, it has been shown that ViT requires considerably fewer computational resources to train compared to computationally demanding CNN backbones such as ResNet. Clearly, the paper has the potential to re-ignite another wave of excitement for exploring this great learning model on different computer vision tasks. To this end, I believe this paper has enough merit to be accepted.

The paper delivers a strong message " transformers can be a more powerful, yet efficient,  compared to the SOTA CNN backbones for image recognition tasks if there is a large-enough dataset available " and the authors prove this claim by performing comprehensive experiments on several large-scale image recognition benchmarks.  This level of experimental verifications is only possible if huge computation re-sources are only available, which is not accessible for most research teams, esp in academia. Otherwise, a similar message (replacing CNN with transformers/attention) has been attempted to be verified in a few earlier works (as acknowledged in this paper as well).

Also, it is hard to imagine that ViT can yet compete with the CNN backbones for vision tasks beyond the considered tasks because:

1- it is not clear how the simple ViT can be extended for the vision tasks which require pixel-level predictions, e.g. image segmentation, depth prediction etc, or 3D vision tasks while being still computationally tractable.

2-  As argued in the paper, transformers are data-hungry to perform as well as ResNet for a task and the availability of such large-scale annotations (e.g. 300M samples) beyond image labelling may not be feasible yet.

3-  Considering transferring a pre-trained model,  there is no evidence that a pre-trained ViT (e.g. on a large amount of data) can still learn a reliable representation for the other basic vision tasks beyond image recognition such as image segmentation or object detection, and still carry a superior performance compared to ResNet when it is fine-tuned... The superiority of DETR (Carion et al 2020) on the detection task is not solely due to the use of a transformer but rather formulating object detection as a set prediction problem to avoid heuristics such as NMS.

Other comments:

I couldn't find any ablation study on different choices of 2D patches sizes versus training time and accuracy except partial experiments on 14,16 and 32. I am not sure what would be best patch size given any image, eg a panoramic image with high resolution when considering both accuracy and training computation compared to ResNet.

---

> ### Author Response · Authors · 2020-11-23
> **Thanks and response to concerns**
>
> Thank you for your detailed review, below we address the key concerns.
>
> > it is not clear how the simple ViT can be extended for the vision tasks which require pixel-level predictions, e.g. image segmentation, depth prediction etc, or 3D vision tasks while being still computationally tractable.
>
> We do not see conceptual blockers to applying ViT models to tasks beyond classification. Similarly to ConvNets, one can augment ViT with specialized heads for predicting dense per-pixel outputs. Processing high-resolution images can be unlocked by efficient transformer variants. However, we note that for ConvNets tackling these non-classification tasks took years of research, so we believe it is fair to leave proper investigation of these topics out of the scope of this paper.
>
> > Transformers are data-hungry <...> and the availability of such large-scale annotations (e.g. 300M samples) beyond image labelling may not be feasible yet. <...> there is no evidence that a pre-trained ViT (e.g. on a large amount of data) can still learn a reliable representation for the other basic vision tasks beyond image recognition such as image segmentation or object detection
>
> We agree that it is an open question whether ViT will transfer to other vision tasks as well as ConvNets do. However, we have preliminary experiments in this direction, which suggest that it might. Therefore, we expect that there won’t be fundamental differences in using ViT instead of ConvNets trained on large scale classification as backbone for other tasks. We leave detailed investigation of this topic for future work.
>
> > I couldn't find any ablation study on different choices of 2D patches sizes versus training time and accuracy except partial experiments on 14,16 and 32.
>
> Figure 8 in the appendix shows how changing different model parameters, including patch size, affects the compute requirements and the performance. Overall, we found that mid-sized patches roughly in the 14-32 pixels range provide the best compute-performance tradeoff.

---

### Official Review · AnonReviewer2 · 2020-10-27
**Transformers for image classification. Strong performance on when pre-trained on large dataset.**

**Rating:** 7
**Confidence:** 4

**Review:**

## Summary
This paper studies to adapt transformer model for image classification task. The new model performs comparably well on various popular benchmarks, and will out-perform when pre-trained on large dataset.

## Pros
1. The technical solution is surprisingly simple, yet achieves strong performance. All it needs is to cut input images into patches, and reshape it as an input sequence. The transformer architecture is kept almost the same as in NLP tasks. The simplicity of the model makes it easy to generalize many vision tasks potentially.
2. The experimental section has clearly demonstrated the pros and cons of the proposed model. The authors not only show the strong performance but also stay upfront about the limits.
3. The additional analysis, visualizations, and self-supervised pre-training sections are very informative.
4. The writing is mostly clear and easy to follow.

## Cons
1. The arguments about "inductive biases" are confusing and self-contradictory. On one hand,  the introduction section says that CNN generalize better due to the inductive biases such as translation equivariance and locality. On the other hand, the rest of the paper claims that avoid inserting inductive biases into the transformer is an advantage.  In my understanding, these inductive biases have been shown to be beneficial for most vision tasks in CNN. Why not introducing some of them into transformer? Please clarify.
2. It's not clear to me how the few shot accuracy is computed. Is the the same evaluation as in self-supervised representation learning? What kind of regularization is applied to the linear regression, and how to learn the mapping?

## Minor
1. It might be better to give reference when mentioning ````100B parameters in the first paragraph of introduction.
2. $z_0^L$ on page 2 bottom seems to be a typo. Should it be $z_L^0$?
3. In Sec.1 ```Hybrid Architecture, the statement "In this hybrid model, the patch embedding projection E (Eq. 1) is replaced by the early stages of a ResNet" is misleading. As in Eq.(1) ${\bf E}$ is applied on different input patches $x_p^i$, whereas early stages of ResNet are applied on the same input (image $x$).

## Questions
1.  If I understand correctly, a model like ViT-L/16 means that it takes 16x16 fixed-sized patches as input. Does this mean that the images get resized to fixed resolution before cutting into patches? How big are those patches? Does the size matter?
2.  As shown in prior work (Unsupervised Visual Representation Learning by Context Prediction, Doersch etal., ICCV 2015), enforcing the patches to be discontinued and with some randomness is beneficial to self-supervised representation learning. Have the authors tried similar processing method? An ablation on different ways of generating patches (discontinued, overlapped, mixed-scale...) might be useful as some simple techniques might greatly improve the performance.

---

> ### Author Response · Authors · 2020-11-23
> **Thanks and response to concerns**
>
> Thank you for your detailed review, below we address the key concerns.
>
> > The arguments about "inductive biases" are confusing and self-contradictory. Introduction says that CNN generalize better due to the inductive biases;  the rest of the paper claims that avoid inserting inductive biases into the transformer is an advantage
>
> As we noted in the introduction, the effectiveness of "inductive biases" depends on the amount of training data. "inductive biases" help generalizing when there's insufficient training data, while it becomes less useful and potentially constraining with sufficient training data. Figure 4 verifies this point of different rankings of ResNet (more inductive bias) and Transformers with different amounts of training data. Therefore, we see no contradiction here.
>
> > It's not clear to me how the few shot accuracy is computed.
>
> We cast few-shot classification as regularized linear least-squares regression. The input features are given by ViT’s penultimate layer, while the target is {-1, 1} encoded vector of labels. This formulation allows us to recover the exact solution in closed-form. Note, that using L2 loss for classification is a known practice, see https://arxiv.org/pdf/2006.07322.pdf.
>
> As a side-note, we do not focus on these few-shot classification as a key metric in this work, since we used it for model selection. The few-shot results of ViT models are very good and we strongly suspect that this is not due to overfitting because of model selection, but we preferred to stay on the safe side and focus on fine-tuning as the key metric.
>
> > If I understand correctly, a model like ViT-L/16 means that it takes 16x16 fixed-sized patches as input. Does this mean that the images get resized to fixed resolution before cutting into patches? How big are those patches? Does the size matter?
>
> We use X to denote patch size (in pixels) for model ViT-L/X. The ViT-L/16 model takes 16 pixels x16 pixels patches as input. For example, we pretrain our models with 224x224 resolution input images, which results in (224/16)*(224/16)=14*14=196 patches in total. Larger patch size leads to smaller number of patches for the same input, meaning less compute and worse results, as can be seen from a comparison between ViT-{B,L}/16 and ViT-{B,L}/32 in Figure 3.
>
> When we resize the input image during fine-tuning, we keep the patch size, so the number of patches increases. ViT generally gracefully handles the changing number of patches, but we need to interpolate the learned position embeddings as described in section 3.2.
>
> > As shown in prior work (Unsupervised Visual Representation Learning by Context Prediction, Doersch etal., ICCV 2015), enforcing the patches to be discontinued and with some randomness is beneficial to self-supervised representation learning. Have the authors tried similar processing method?
>
> We thank the reviewer for the pointer. This might indeed benefit our preliminary results on self-supervised representation learning. However, because an in-depth study on self-supervised representation learning is beyond the scope of this paper we would like to defer this to future work. Note that this is not relevant for supervised pre-training.
>
> > Minor
>
> Thanks, corrected

---

### Official Review · AnonReviewer1 · 2020-10-28
**It is a good idea. More importantly, it is great to make the idea work.**

**Rating:** 7
**Confidence:** 4

**Review:**

This paper explores the Transformer architecture for image recognition. The authors mainly follow the settings of BERT in NLP where the input is a sequence of words. To adopt Transformer for image recognition, the authors split the image into grid patches as inputs for the vision Transformer (ViT). Other settings like adding a special token at the beginning for classification and adding position embeddings are the same as those used in BERT. While Transformer is not new, the way it is applied on raw input images directly for image classification has not been done before. I also like the analogy with the hybrid model where the early layers in ResNet serve the role similar to the patch embedding projection. More importantly, the authors have conducted extensive experiments to validate the effectiveness of the proposed approach. Some of the visualizations are also insightful. The main conclusion is that ViT can outperform BiT(ResNet) when pre-trained on large-scale training data with supervised training.

As the Transformer model has become more popular in the vision community, it seems natural for the authors to come up with ViT for image recognition. I think the main contribution of this paper is not to propose this idea, but rather to make it work. The fact that ViT can outperform BiT with ResNet only when it scales-up to a huge model trained on a huge dataset makes it impossible for most researchers with limited resources to conduct such experiments. Although the results may not be that encouraging in my opinion, I think the experiments conducted in this paper are valuable for the research community.

My major concern is that the pre-training conducted in this paper is fully supervised. This is different to BERT that can use unlabeled text data or Vision-Language Pre-training where weakly-labeled image-text pairs can be leveraged. The authors briefly mentioned that the masked patch prediction can still improve ImageNet classification accuracy compared to training from scratch, but it has a gap with supervised pre-training. It would be great to explore self-supervised pre-training in the future as also pointed out by the authors.

Another concern is about the comparison with the hybrid model. The main drawback of applying ViT on the full image is that it cannot scale to large input resolution due to memory constraint. A hybrid approach seems to be a good work-around to handle this case. While the authors have mentioned the hybrid model, very limited comparisons are conducted in the experiments. Only Figure 5 shows some results of hybrid models. In some cases, the hybrid model outperforms ViT. Why not add more data points for hybrid models at higher pre-training compute cost? Why not compare the numbers in Table 2 and Figure 2-4?

In the appendix, the authors mention that the absolute numbers of BiT are lower than those reported by Kolesnikov et al. (2020), since they pre-train only for 7 epochs, not 30. Is it possible that there is more room to improve for BiT (ResNet) in Figure 5 if we increase the pre-training cost? At least the trend does not show saturating performance of BiT.

Overall, I think the authors have done a great job in conducting all these experiments. Although the results are not that encouraging as these experiments require too many resources, I think the experiments can provide some value for future research. I would like the authors to comment on the comparison with BiT and the hybrid model in Figure 5.

---

> ### Author Response · Authors · 2020-11-23
> **Thanks and response to concerns**
>
> Thank you for your detailed review, below we address the key concerns.
>
> > “The fact that ViT can outperform BiT with ResNet only when it scales-up to a huge model trained on a huge dataset makes it impossible for most researchers with limited resources to conduct such experiments.”
>
> We agree that upstream pre-training of the largest models requires substantial computational and data resources. However, smaller models (B/32, B/16) pre-trained on the smaller ImageNet-21k dataset have more modest computational requirements and still perform very well. Moreover, we also would like to point out that the ViT model needs to be pre-trained once, then their fine-tuning on a custom dataset of interest can be done with modest computational resources in a data-efficient manner. After anonymity requirements are lifted, we will publish the link to pretrained models and code.
>
> >  “It would be great to explore self-supervised pre-training in the future as also pointed out by the authors.”
>
> We fully agree. However, extensive self-supervised experiments would increase the scope of this paper beyond 8-page conference submission, thus, we leave it for future research. We hope that open-source release of the code and models will facilitate the progress in this direction.
>
> > Another concern is about the comparison with the hybrid model. <...> Why not add more data points for hybrid models at higher pre-training compute cost? Why not compare the numbers in Table 2 and Figure 2-4?
>
> We did experiment with hybrid models in the initial stages of the project and (to our big surprise) found out that the benefits of having an initial ConvNet stage in these models wash out if we scale up the transformer part of the architecture. Figure 5 confirms this observation, and to further verify it, we added one even larger hybrid model in the revised version. Because of this, and since we were interested in scalability of architectures, we opted to focus more on the surprising efficiency of pure transformers in the main experimental section. However, in applications where smaller models are of interest hybrid can be an excellent choice and further investigation of their properties is an exciting direction for future work. Finally, please note that we have added many numeric results for hybrid models in Table 6 in the Appendix.
>
> > Is it possible that there is more room to improve for BiT (ResNet) in Figure 5 if we increase the pre-training cost?
>
> Figure 5 studies the trade-off between compute and model accuracy. As you note, increasing the amount of compute (number of epochs) would result in better performance, which is uniformly true for BiT, Hybrid and ViT models. The main question Figure 5 answers is “which class of models has better scaling properties?”. And the answer seems to suggest that Transformer models offer better scaling than BiT models. Hybrids are better than pure transformers at smaller model sizes, but this advantage vanishes for larger models.
>
> We added two even larger ResNets to Figure 5 to better highlight the scaling trend and they fit the previously observed trend. Another comparison point is provided by Table 2: ViT-H/14 model, which was trained for 14 epochs and with substantially less compute overall, outperforms the BiT-L model trained for 40 epochs (although note that this is a less apples-to-apples comparison than in Figure 5).

---

### Official Review · AnonReviewer3 · 2020-11-01
**Good paper with interesting empirical results, but missing some analysis**

**Rating:** 7
**Confidence:** 5

**Review:**

This paper introduces a Transformer-based image recognition model that is fully built on the Transformer layers (multi-head self-attention + point-wise MLP) without any standard convolution layers. Basically, it splits an image into patches and takes as input the set of linear embeddings of the patches and their positions. For classification, a learnable class token is added to the input and a classification head (MLP) is attached to the class token output of the final Transformer layer. Extensive experiments of transfer learning show that when pretrained on a sufficiently large dataset (100~300M), the proposed Vision Transformer can outperform state-of-the-art convolutional networks with less training cost as well as less number of parameters.

Pros
1. Clearly motivated and well written
- The background of the research, the motivation of Vision Transformer, and the related work are all clearly stated and summarized.

2. The first fully-Transformer-based image recognition model
- The simple yet effective Vision Transformer is introduced by adapting the original Transformer with minimal modification; patch embedding, class token, and class head.

3. Extensive evaluation and good analysis
- This paper demonstrates the power of the Vision Transformer model by extensive large-scale experiments, outperforming SOTA CNN models. Comparative evaluation with important baselines, ResNet and hybrid models, are well designed and conducted with different scales of datasets and models. The results are impressive and interesting. Additional discussions in Appendix are also useful in understanding the model.

Cons
1. No significant technical novelty
-  The proposed model is incremental modifications of the original Transformer and its existing variants.

2. Lack of further analysis of the inductive bias
- The authors contrast the Transformer with CNN, stating “Transformers lack some inductive biases inherent to CNNs, such as translation equivariance and locality, and therefore do not generalize well when trained on insufficient amounts of data.” and “the convolutional inductive bias is useful for smaller datasets, but for larger ones, learning the relevant patterns is sufficient, even beneficial.” However, it is not clear which inductive bias of CNN prevents better generalization; is it translation equivariance, locality, static weights of kernels, or static size of kernels? I don’t find any clear answers from the paper. Since there exist different inductive biases in standard convolution, it may be the case that some of them help generalization and some others do not; worse generalization of CNN with a larger dataset may be affected by all those factors in a tangle. In this regard, further analysis of this issue would improve this work. E.g., in order to check the benefit of locality bias, ViT with local self-attention can also be compared, etc.

3. Some misleading statements
- The authors repeatedly call the input of patch embeddings an “input sequence”, which I guess is intended to remind of the NLP origin. But, this may confuse readers. The Transformer used in this work does not process the input as a sequence, and actually the output is equivariant to any permutation of the patch (+position) embeddings.
- “Transformers lack some inductive biases inherent to CNNs, such as translation equivariance and locality”. This needs to be clarified. As I noted above, since the Transformer in this work is permutation equivariant, it can also be seen as translation equivariant. As for “locality”, the Transformer also performs point-wise (patch-wise) processing, thus leveraging locality.
- “Unlike prior works using self-attention in computer vision, we do not introduce any image-specific inductive biases into the architecture.” This statement in conclusion is an overclaim, conflicting with the statement at the end of Sec. 2.1 (resolution adjustment and patch extraction).
- The title of this paper “An image is worth 16x16 words”, what does it mean? I don’t find an answer from the paper.

---

> ### Author Response · Authors · 2020-11-23
> **thanks and response to concerns**
>
> Thank you for your detailed review, below we address the key concerns.
>
> > Lack of further analysis of the inductive bias
>
> We agree that a more in-depth analysis of inductive bias would be a very interesting topic to study. Although not exhaustive we conducted multiple experiments that give interesting insights into the role of inductive bias:
> One is the comparison between Transformers, ResNets and hybrids in Figure 5. We find that in the large model regime (the right side of the plot) with enough data, the impact of adding inductive bias by combining ViT with ResNet backbones (i.e., our hybrids) vanishes.
> We find that ResNets which have arguably more inductive bias perform very well in the smaller data regimes (Figure 3, 4), whereas comparable transformers require much more data to achieve similar and eventually even better results.
> Adding more image-specific inductive bias in the form of more structured positional embeddings (Appendix D.3) did not make any difference in the high data regime.
> All these findings lead us to the conclusion that with growing models and datasets, image-specific inductive biases becomes less important.
>
> > The use of “input sequence” is not appropriate, since transformers operate on sets
>
> We believe this is a matter of perspective. We refer to the embedded input patches *without positional embeddings* as the input sequence. Because of this, the order of those patches actually matters.
>
> > “Transformers lack some inductive biases inherent to CNNs, such as translation equivariance and locality”. This needs to be clarified.
>
> We agree that this statement is only partially true and needs clarification. We added a subsection on this at the end of Section 3.1. Locality is somewhat preserved by our model, but only within patches and only in MLP layers, while self-attention layers are global. Concerning translation equivariance, arbitrary translations of the input will not result in corresponding translations of the output because of the learnable position embeddings that are added to the patches and stay fixed when the input is translated.
>
> > “Unlike prior works using self-attention in computer vision, we do not introduce any image-specific inductive biases into the architecture.” This statement in conclusion is an overclaim, conflicting with the statement at the end of Sec. 2.1
>
> We agree that patch extraction is an image-specific inductive bias, so we will soften this claim; the only image-specific bias is in this initial step. Resolution adjustment is not part of the architecture itself, so we don’t see a conflict.
>
> > The title of this paper “An image is worth 16x16 words”, what does it mean?
>
> This is merely a wordplay based on the fact that our largest model (H/14), when trained on the standard ImageNet resolution 224x224 pixels, splits the input image into 16x16=256 patches, and we feed these patches to a transformer in the same way words are fed to transformers in NLP.

---

### Public Comment · ~Sachin_Mehta1 · 2020-11-12
**Interesting paper; missing discussion about BoW and MIL approaches; Positional encoding**

The paper does not discuss bag of visual words (BoW)/multi-instance learning (MIL) approaches. These approaches also splits an image (bag) into words (patches or instances) and then learn representations on words, similar to what this paper does. Though current CNN-based models do not use BoW/MIL approaches for standard computer vision tasks (classification, detection, and segmentation), they are widely used in histopathological images [R1]. Most of these also use attention (e.g., [R2]).

Also, I do not understand the need of positional encoding for image classification. In NLP, it makes sense because the same token can appear in multiple places in a sentence. But for images, patch order is fixed.  Another parallel work [R3] that uses Transformers for classifying histopathological images (order of GigaPixels) shows that with transformers with CNNs (similar to hybrid model in ViT), networks can learn clinically relevant biomarkers without any positional encoding. I would appreciate if you can elaborate more on the effect of positional encoding.

[R1] Hou, Le, et al. "Patch-based convolutional neural network for whole slide tissue image classification." CVPR. 2016.

[R2] Maximilian Ilse, Jakub Tomczak, Max Welling ; ICML, 2018.

[R3] https://arxiv.org/pdf/2007.13007.pdf

---

> ### Author Response · Authors · 2020-11-18
> **ViT != BoW**
>
> Regarding relation to early BoW models we note that our model is conceptually quite different from those models, as ViT models interaction between all patches throughout the whole network through global self-attention layers.
>
> Positional embeddings are crucial to incorporate information about relative patch locations, as otherwise the output of the ViT model will be invariant to any patch permutation. In the appendix we ablate positional embeddings and demonstrate that they significantly boost the performance.

---

### Public Comment · ~Louis_THIRY1 · 2020-11-16
**Performance as "Bag-of-patches" method (i.e. without positional embedding)**

Dear authors,

Thanks for this very interesting work.

According to table 3, the base model with 16x16 patches  ViT-B/16 pre-trained on ImageNet has a Top-1 accuracy around 78 %. According to the ablation study shown in table 7, the gap between a model with positional embedding and without positional embeding (= bag-of-patches) is "relatively small" (3% out of 64 %).

Do you think that such a "relatively small" gap  would hold for a ViT-B/16 model pre-trained on ImageNet ? Do you expect the "bag-of-patches" version of  the model ViT-B/16  pre-trained on ImageNet to have a Top1 accuracy greater than 70% ?

If this is the case,  your model compares very favorably with a BagNet using similar patch-size ( Bag-of-patches ResNet50 , see the ICLR 2019 publication https://openreview.net/pdf?id=SkfMWhAqYQ ). With patch-size 17, a BagNet with a ResNet50 backbone has 58.8 Top1 Acc and 81.2 Top5 accuracy. Then your model is more powerfull than ResNets to classify the images based on patches/textures. This would also suggest that images can be very well classified as a Bag-of-small-patches and that ability to recognize shapes is not necessary for a high performance image classification pipeline.

Thanks in advance,

Louis THIRY

---

> ### Author Response · Authors · 2020-11-18
> **ViT without positional embeddings != BagNet**
>
> Ablating positional embeddings on ImageNet is an interesting experiment, we may try running it soon.
>
> Note, however, that the BagNet model is conceptually different from the ViT model without positional embeddings. It is true that both models are invariant to patch order, but, unlike BagNet, ViT does model interactions between all patches and thus has much more powerful modeling capabilities.

---

### Public Comment · ~Martin_Jaggi1 · 2020-11-17
**Novelty over (Cordonnier et al., 2020)**

There is a confusion in the discussion of (Cordonnier et al., 2020), whose approach matches ViT-Small/2 in your terminology.
(Cordonnier et al., 2020) should not be termed *local*, but uses *global* self-attention on the full size images by extracting 16x16 patches of size 2x2 to reduce memory footprint.

We would propose to remove “We are not aware of prior application of Transformers with global self-attention to full-sized images” and to replace

> Such local multi-head dot-product self attention blocks can completely replace convolutions (Ramachandran et al., 2019; Cordonnier et al., 2020; Zhao et al., 2020). Alternatively, works such as Sparse Transformers (Child et al., 2019) employ scalable approximations to global self-attention in order to be applicable to images.

by

> Such local multi-head dot-product self attention blocks can completely replace convolutions (Ramachandran et al., 2019; Zhao et al., 2020). Alternatively, (Cordonnier, et al. 2020) extract patches of size 2\times 2 on the full size images to limit the memory footprint. Sparse Transformers...

---

> ### Author Response · Authors · 2020-11-18
> **thanks, will correct**
>
> Thanks for pointing this out, we will make sure to adjust the discussion properly in the updated version of the paper.

---

> ### Author Response · Authors · 2020-11-19
> **suggested edit**
>
> Dear Martin,
>
> Thanks for bringing this up! We are sorry we haven't described your work properly in the deadline rush. Thinking about it, it seems the work warrants a stronger highlight than what you suggested. What do you think about the following option?
>
> <...>
> \citet{parmar18-imagetransformer} applied the self-attention only in local neighborhoods for each query pixel instead of globally.
> Such local multi-head dot-product self attention blocks can completely replace convolutions \citep{ramachandran19-sasa,zhao2020-san}. In a different line of work, Sparse Transformers \citep{child2019-sparsetransformers} employ scalable approximations to global self-attention in order to be applicable to images. <...> Many of these specialized attention architectures demonstrate promising results on computer vision tasks, but require complex engineering to be implemented efficiently on hardware accelerators.
>
> Most related to ours is the model of~\citet{cordonnier2020-sacnn}, which extracts patches of size $2 \times 2$ from the input image and applies full self-attention on top. This model is very similar to ViT, but our work goes further to demonstrate that large scale pre-training makes vanilla transformers competitive with (or even better than) state-of-the-art CNNs. Moreover, \citet{cordonnier2020-sacnn} use a small patch size of $2 \times 2$ pixels, which makes the model applicable only to small-resolution images, while we handle medium-resolution images as well.
>
> <...>
>
> Another recent related model is image GPT (iGPT)~\citep{chen20-igpt}, which applies Transformers <...>

---

### Author Response · Authors · 2020-11-23
**Thanks for the reviews and summary of key paper changes**

We thank the reviewers for their thoughtful reviews. The reviewers appreciated the combination of the method’s simplicity and good performance (R2, R3, R4), comprehensive experimental evaluation (R1, R2, R3, R4), and good presentation (R2, R3).

Since the concerns voiced by the reviewers are mainly non-overlapping, we provide detailed responses in individual responses to each review. Below we summarize the key changes made to the paper:
- Added a discussion of inductive biases in section 3.1.
- Added two larger ResNets and one larger hybrid architecture to Figure 5 (scaling of different architectures). The trends are as before, but now visible even clearer.
- Adjusted the discussion of related work of Cordonnier et al., added several more related works.
- Added an ImageNet-21K-pretrained model to Table 2. It is somewhat below SOTA, but still performs very well and takes less compute to pre-train.
- Slightly improved the fine-tuning results on ImageNet and VTAB by better tuning the hyperparameters.
- Added a table with detailed results of scaling experiments (Table 6).
- Added additional technical details and polished the text throughout .

---

### Comment · ~Fajie_Yuan1 · 2021-01-24
**a small question, thanks**

Hi,

thanks for your great work. I wonder why not perform pertaining using the autoregressive language  model (LM) or masked LM like GPT and Bert pertaining.  Should we compute the similarity between the input embedding and output hidden using some loss?
I am afraid Just adding a linear projection layer at. the beginning may not help learn the best representation.

---

> ### Author Response · Authors · 2021-01-24
> **self-supervised ViT makes sense**
>
> Hi,
>
> Self-supervised pre-training makes sense, in fact we gave it a quick try and report some results in the paper (sec. 4.6). We did not push it further, since in our experience supervised training typically allows for better performance with the same amount of compute. However, given quite promising representation learning results from iGPT, it's reasonable to assume that one could get good results with self-supervised ViT too. How to do it best is a matter of future research.

---

### Decision · Program_Chairs · 2021-01-07
**Final Decision**

**Decision:**

Accept (Oral)

**Comment:**

This paper has generated a lot of great discussion and it presents a very different way of doing image recognition at scale compared to current state of the art practices.  All reviewers rated this paper as an accept.
This work is interesting enough that in my view it really deservers further exposure and discussion and an oral presentation at ICLR would be a good way to achieve that.